

# Jatropha curcas ortholog of tomato MADS-box gene 6 (JcTM6) promoter exhibits floral-specific activity in Arabidopsis thaliana

Jing-Xian Wang[1,2], Xin Ming[1,2], Yan-Bin Tao[2,3] and Zeng-Fu Xu[2,3]

[1] School of Life Sciences, University of Science and Technology of China, Hefei, Anhui, China
[2] CAS Key Laboratory of Tropical Plant Resources and Sustainable Use, Xishuangbanna Tropical Botanical Garden, Innovation Academy for Seed Design, Chinese Academy of Sciences, Menglun, Mengla, Yunnan, China
[3] Center of Economic Botany, Core Botanical Gardens, Chinese Academy of Sciences, Menglun, Mengla, Yunnan, China

## ABSTRACT

**Background**. *Jatropha curcas* L., a perennial oilseed plant, is considered as a promising feedstock for biodiesel production. Genetic modification of flowering characteristics is critical for *Jatropha* breeding. However, analysis of floral-specific promoters in *Jatropha* is limited.

**Methods**. In this study, we isolated the *Jatropha* ortholog of *TM6* (*JcTM6*) gene from *Jatropha* flower cDNA library and detected the expression pattern of *JcTM6* gene by quantitative reverse transcription-polymerase chain reaction (qRT-PCR). We isolated a 1.8-kb fragment from the 5' region of the *JcTM6* gene and evaluated its spatiotemporal expression pattern in *Arabidopsis* using the *β-glucuronidase* (*GUS*) reporter gene and *Arabidopsis ATP/ADP isopentenyltransferase 4* (*AtIPT4*) gene, respectively.

**Results**. *JcTM6* was identified as a flower-specific gene in *Jatropha*. As expected, *JcTM6* promoter was only active in transgenic *Arabidopsis* flowers with the strongest activity in stamens. Moreover, *JcTM6:AtIPT4* transgenic *Arabidopsis* showed a phenotype of large flowers without any alterations in other organs. Furthermore, deletion of the region from −1,717 to −876 bp resulted in the disappearance of promoter activity in stamens but an increase in promoter activity in young leaves, sepals, and petals. Deletion analysis suggests that the −1,717- to −876-bp promoter fragment contains regulatory elements that confer promoter activity in stamens and inhibit activity in young leaves, sepals, and petals.

## INTRODUCTION

Promoter plays a significant role in gene expression regulation. Three types of promoters are currently employed in plant genetic engineering: constitutive, tissue-specific, and inducible promoters (*Muthusamy et al., 2017*; *Potenza, Aleman & Sengupta-Gopalan, 2004*). Tissue-specific promoters drive transgene expression in a specific spatiotemporal pattern, which is effective in the modification of agronomic traits of crop plants. For example, the rice (*Oryza*

Corresponding authors
Yan-Bin Tao, taoyanbin@xtbg.ac.cn
Zeng-Fu Xu, zfxu@xtbg.ac.cn

*sativa* L.) gene *OsGA2ox1* encodes a gibberellin (GA) catabolic enzyme, GA 2-oxidase (*Lester et al., 1999*; *Martin, Proebsting & Hedden, 1999*; *Thomas, Phillips & Hedden, 1999*). When the expression of *OsGA2ox1* was driven by the constitutive *Actin* promoter, transgenic rice plants failed to set grains. To prevent sterility, the promoter of a GA biosynthesis gene, *OsGA3ox2*, which encodes GA 3-oxidase and is specifically active in shoots, was used to control the expression of *OsGA2ox1*. As expected, transgenic rice exhibited a semi-dwarf phenotype with normal yield (*Sakamoto et al., 2003*). GA 20-oxidase is a GA biosynthetic enzyme in plants (*Coles et al., 1999*). In poplar (*Populus* spp.), overexpression of the *Pinus densiflora GA 20-oxidase* gene (*PdGA20ox*) under the control of the constitutive *35S* promoter increased GA levels, thereby accelerating stem growth and plant biomass; however, transgenic poplar plants showed poor leaf development and root growth. When the *PdGA20ox* gene was driven by a xylem-specific promoter *DX15* from poplar, the undesirable phenotypes were reduced (*Jeon et al., 2016*).

Physic nut (*Jatropha curcas* L.) is an oilseed plant belonging to the Euphorbiaceae family. The seed oil of *Jatropha* is a promising feedstock for biodiesel production (*Kumar & Sharma, 2008*). However, low seed yield, which is mainly caused by low female: male ratio, is a long-standing problem in *Jatropha* (*Raju & Ezradanam, 2002*; *Rao et al., 2008*). *Jatropha* is a monoecious plant species with male and female flowers on the same inflorescence, and the average ratio of female to male flowers is 1:13−1:29 (*Raju & Ezradanam, 2002*; *Tewari et al., 2007*). There are 100−300 flowers in each inflorescence of *Jatropha*, which only produce approximately 10 fruits (*Kumar & Sharma, 2008*; *Pan & Xu, 2011*). Hence, genetic modification of flowering characteristics is critical for *Jatropha* breeding. Floral-specific promoters play crucial roles in this modification because they can drive efficient expression of functional genes in flowers without affecting the vegetative growth of plants. In pea (*Pisum sativum*), the *PsEND1* promoter exhibits anther-specific activity. Expression of the ribonuclease gene *barnase* (*Gardner, Felsheim & Smith, 2009*) in *Arabidopsis* and *Brassica napus* under the control of the *PsEND1* promoter causes anther ablation at an early developmental stage, leading to male sterility (*Roque et al., 2007*). *Arabidopsis APETALA3* (*AP3*) promoter was identified as a floral-specific promoter in petunia (*Petunia x hybrida*). Expression of the *Agrobacterium tumefaciens isopentenyltransferase* (*ipt*) gene under the control of the *AtAP3* promoter in petunia increased the flower size, without affecting vegetative development (*Verdonk et al., 2008*). However, analysis of promoters, especially floral-specific promoters, in *Jatropha* is limited. Although the *Jatropha APETALA1* (*JcAP1*) promoter was recently identified as a reproductive tissue-specific promoter showing high activity in inflorescence buds and seeds (*Tao et al., 2016*), it is not sufficient to address transgene expression analysis in *Jatropha*.

In this study, we isolated the promoter of the *Jatropha* ortholog of *TOMATO MADS-BOX GENE 6* (*JcTM6*), a floral-specific gene. The activity of *JcTM6* promoter was evaluated in *Arabidopsis* using the *β-glucuronidase* (*GUS*) reporter gene. The results of GUS staining showed that the *JcTM6* promoter was active only in flowers, with the highest activity in stamens. By using this promoter directed a cytokinin biosynthesis gene, *Arabidopsis ATP/ADP isopentenyltransferase 4* (*AtIPT4*) gene (*Li et al., 2010*), only flower phenotype was changed in transgenic *Arabidopsis*. Furthermore, deletion analysis showed that an

approximately 0.85-kb fragment of the *JcTM6* promoter (−1717 to −876 bp) is critical for maintaining its floral-specific expression pattern.

## MATERIALS & METHODS

### Plant materials

Plants of *Jatropha curcas* and *Arabidopsis thaliana* ecotype Columbia (Col-0) were used in this study. *Jatropha* plants were cultivated in Xishuangbanna, Yunnan Province, China, as described previously (*Pan & Xu, 2011*). *Arabidopsis* plants were grown in an environmentally controlled room at 22 °C under 16-h light/8-h dark photoperiod.

### *JcTM6* expression analysis

The *JcTM6* gene (GenBank accession no. MN820724) was identified in the *Jatropha* flower cDNA library (*Chen et al., 2014*). Quantitative reverse transcription-polymerase chain reaction (qRT-PCR) was performed to examine the expression level of *JcTM6* in different organs of *Jatropha* (roots, stems, young leaves, mature leaves, inflorescence buds, female flowers, male flowers, pericarps and seeds at 42 days after pollination (DAP), male sepals and petals, stamens, female sepals and petals, and pistils) and *Arabidopsis* (leaves and flowers). Total RNA from each organ was isolated using the silica particle extraction method (*Ding et al., 2008*). Then, qRT-PCR was performed as previously described in *Tao et al. (2015)*. The *JcGAPDH* and *AtActin* were used as an internal control for data normalization. Primers used for qRT-PCR are listed in Table 1. The results of qRT-PCR were obtained from three biological replicates and three technical replicates.

### Cloning of the upstream region of *JcTM6*

The 5′ region of *JcTM6* was isolated from *Jatropha* genomic DNA by genome walking (*Siebert et al., 1995*) according to the Genome Walker™ Kit Universal User Manual (Clontech). Then, the full-length *JcTM6* promoter was amplified using the primers, XT405 and XT408. The PCR product was cloned into the pGEM-T Easy vector. Putative cis-acting elements in the *JcTM6* promoter were analyzed using the PLACE database (*Higo et al., 1999*). The transcriptional start site of *JcTM6* was identified as previously described in *Tao et al. (2016)*. Primers employed for genome walking and 5′-RACE are listed in Table 1.

### Construction of *JcTM6* promoter-GUS fusion and *Arabidopsis* transformation

To generate the JcTM6:GUS plasmid, *Xba* I and *Bam* HI were used to digested pBI101 (*Jefferson, Kavanagh & Bevan, 1987*), and the pGEM -T Easy vector containing the *JcTM6* promoter, respectively. The resulting fragments were ligated using the T4 DNA Ligase (Promega) to generate the *JcTM6:GUS* fusion construct. Then, the *JcTM6:GUS* plasmid was introduced into *Agrobacterium tumefaciens* EHA105 by electroporation (GenePulser Xcell; Bio-Rad), and the transformed *A. tumefaciens* cells were used to transform *Arabidopsis* plants by the floral dip method (*Clough & Bent, 1998*).

### Histochemical GUS staining assay

To perform GUS staining, various tissues of transgenic *Arabidopsis* were submerged in the GUS assay buffer (50 mM sodium phosphate [pH 7.0], 0.5 mM $K_3Fe (CN)_6$, 0.5 mM
**Table 1  Sequences of the primers used in this study.**

| Name | Sequence (5′ to 3′) | Feature |
|---|---|---|
| GSP1 | CTCTTGGAATAAGTAACCTGTCTGTTGG | *JcTM6* gene-specific primer for genome walking |
| GSP2 | CAAAACCCACTACTACAAAACCGAAGA | *JcTM6* gene-specific primer for genome walking |
| XT95 | GCTGCTAAGGCTGTTGGGAA | *JcGAPDH* gene primer for qRT-PCR |
| XT96 | GACATAGCCCAATATTCCCTTCAG | *JcGAPDH* gene primer for qRT-PCR |
| XK712 | TATCTCTTCGGTTTTGTAGTAGTGGG | *JcTM6* gene primer for qRT-PCR |
| XK713 | TCTCTTGGAATAAGTAACCTGTCTGT | *JcTM6* gene primer for qRT-PCR |
| XT405 | TGCTCTAGAAATAGCTATAAAATCAATT | For cloning the full-length promoter and construction of *JcTM6:GUS* |
| XT408 | CGCGGATCCTTTTCCTTTCTTCTTGATA | For cloning the full-length promoter and construction of *JcTM6:GUS* |
| XD548 | GCTCTAGACGCTTACAGAATTTGCGA | For construction of *D:GUS* |
| XB994 | CAATCTTTCCACGACCCATTTTTCCTT | *JcTM6* gene-specific primer for 5′-RACE |
| XK718 | TGTGCCAATCTACGAGGGTTT | *AtActin* gene primer for qRT-PCR |
| XK719 | TTTCCCGCTCTGCTGTTGT | *AtActin* gene primer for qRT-PCR |
| XK984 | TCGCTGAGTTCCACCGCTCTAAG | *AtIPT4* gene primer for qRT-PCR |
| XK985 | AGGGTCCCATTTATCCATGTCATTG | *AtIPT4* gene primer for qRT-PCR |
| XE815 | CCTTGTCAATGGCAAGAAGAGGCAA | *AHK2* gene primer for qRT-PCR (*Nishimura et al., 2004*) |
| XE816 | CACCTTCTGCAACTCGTCTGTT | *AHK2* gene primer for qRT-PCR |
| XE819 | TCAGAGAACATCTTGCCTCGT | *ARR5* gene primer for qRT-PCR |
| XE820 | AGCTGCGAGTAGATATCATTAGCTT | *ARR5* gene primer for qRT-PCR |

$K_4Fe(CN)_6 \cdot 3H_2O$, 0.5% Triton X-100, and 1 mM X-Gluc) and vacuum-infiltrated for 15 min. Then, tissues were incubated overnight at 37 °C, cleared in 70% ethanol (*Jefferson, Kavanagh & Bevan, 1987*), and examined under a stereomicroscope (Leica M80). The results of GUS staining were obtained from five biological replicates and three technical replicates.

## RESULTS

### *JcTM6* expression in *Jatropha*

We identified the *JcTM6* cDNA (GenBank accession no. MN820724) from our *Jatropha* flower cDNA library constructed previously (*Chen et al., 2014*). *JcTM6* encodes a 230-amino acid protein, which shows high similarity to *TM6* homologs from other plant species (Fig. 1A). Phylogenetic analyses showed that *JcTM6*, which contains the paleoAP3 motif, belongs to the *TM6* group, rather than the *euAP3* group (Fig. 1B).

To analyze the expression pattern of *JcTM6* in *Jatropha*, qRT-PCR was performed using total RNA extracted from various tissues including roots, stems, leaves, inflorescences, female and male flowers, and pericarps and seeds at 42 DAP. The *JcTM6* gene was predominantly expressed in female and male flowers (Fig. 2), indicating that *JcTM6* is a flower-specific gene. Furthermore, *JcTM6* showed high expression in the stamens of male flowers and petals of male and female flowers but low expression in sepals and pistils (Fig. 2). Thus, the expression pattern of *JcTM6* in floral organs is consistent with that of class B genes (*Weigel & Meyerowitz, 1994*).

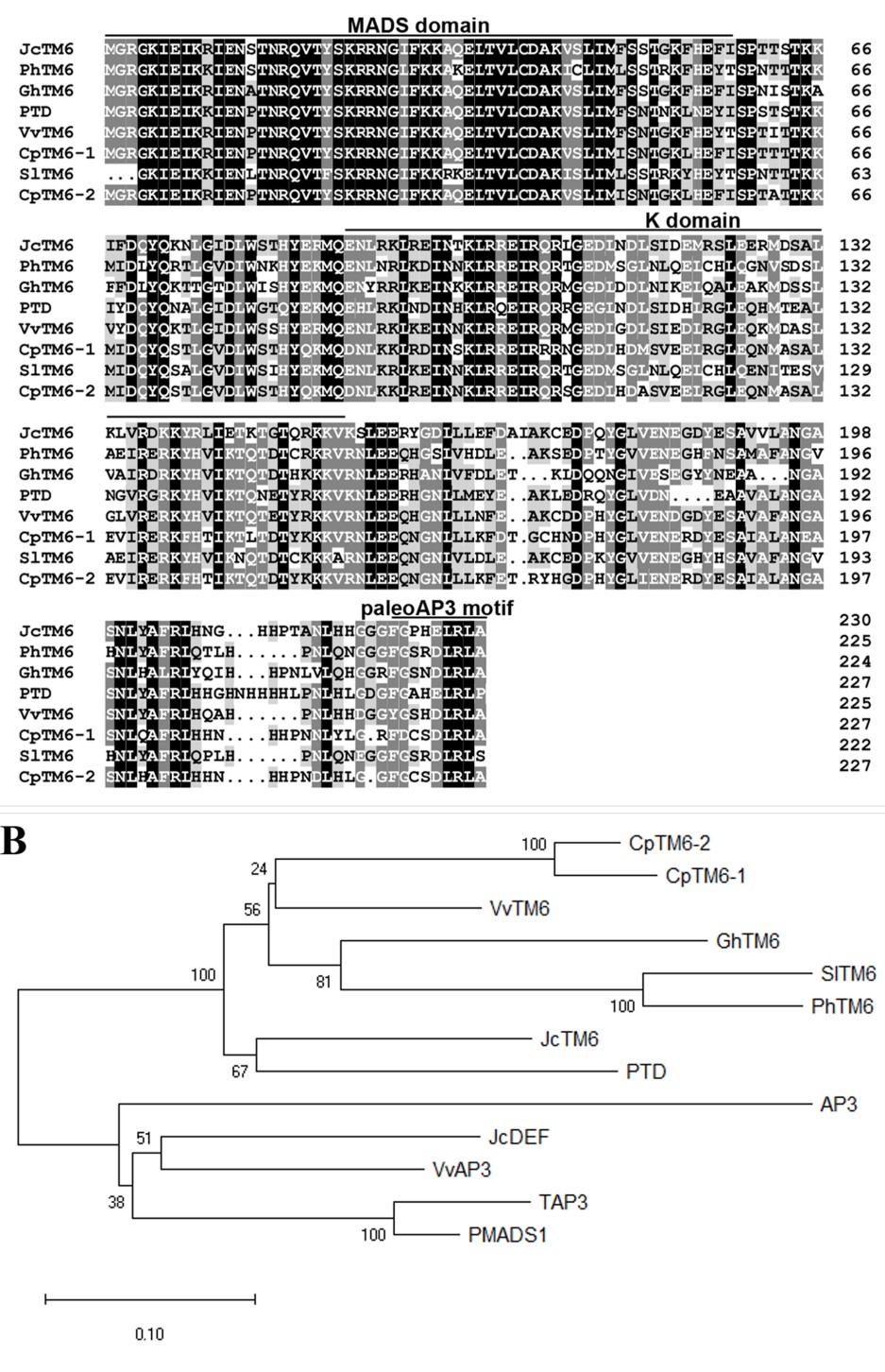

**Figure 1  A comparison of JcTM6 and its homologs.** (A) The alignment of the deduced amino acid sequences of JcTM6 with that of *Vitis vinifera* VvTM6 (accession number DQ979341), *Carica papaya* CpTM6-1 (accession number ABQ51321), and CpTM6-2 (accession number ABQ51322), *Populus trichocarpa* PTD (accession number AAC13695) , *Gossypium hirsutum* GhTM6 

**Figure 1 (…continued)**
(accession number ADX60056), *Petunia x hybrida* PhTM6 (accession number AF230704) and *Solanum lycopersicum* SlTM6 (accession number CAA43171) . Identically and partially conserved amino acid sequences are shown in black and gray, respectively. The conserved regions, MADS domain and K domain and *paleoAP3* C-terminal motif in JcTM6 are underlined. (B) A phylogenetic analysis of JcTM6 and other homologs. *Jatropha curcas* JcDEF (accession number XP_012071964), *Solanum lycopersicum* TAP3 ( accession number ABG73412), *Vitis vinifera* VvAP3 (accession number NP_001267960), *Arabidopsis thaliana* AP3 (accession number BAA04665), *Petunia hybrida* PMADS1 (accession number Q07472). The tree was constructed using MEGA 7.0 software and the neighbor-joining (N–J) method. The N-J unrooted dendrogram was generated from an alignment of the deduced amino acids with the ClustalW program. One thousand replicates were used for the Bootstrap test. The scale bar indicates the average number of substitutions per site.

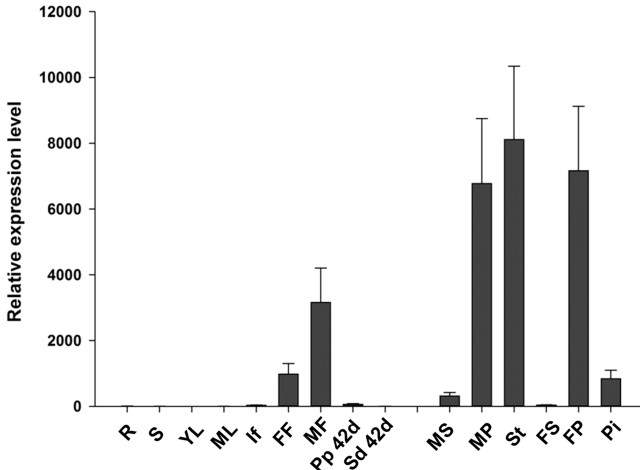

**Figure 2** **Expression pattern of *JcTM6* in *Jatropha*.** Samples from adult plants: roots (R), stems (S), young leaves (YL), mature leaves (ML), inflorescence buds (If), female flowers (FF), male flowers (MF), pericarps at 42 days after pollination (DAP) (Pp 42d), seeds at 42 DAP (Sd 42d), male sepals (MS), male petals (MP), stamens (St), female sepals (FS), female petals (FP), and pistils (Pi). qRT-PCR results were obtained from three biological replicates. The errors denote the SD. The values were normalized to the expression of JcGAPDH (*Zhang et al., 2013*). The relative expression level of young leaves was set as the standard value of 1.

## Isolation and sequence analysis of *JcTM6* promoter

A 1.8-kb fragment of the *JcTM6* promoter (Fig. 3A, −1717 to +103 bp; GenBank accession no. MN044579) was isolated from *Jatropha* genomic DNA by genome walking (*Siebert et al., 1995*). The transcription start site of *JcTM6* was located 103 nt upstream of the translation start codon (Fig. 3A). Analysis of the *JcTM6* promoter using the PLACE database (*Higo et al., 1999*) revealed various putative cis-elements in the 1.8-kb *JcTM6* promoter fragment (Fig. 3A) including two CArG boxes, which act as binding sites for MADS-box transcription factors (*Irish & Yamamoto, 1995*), some pollen-specific elements, including five GTGANTG10 motifs (GTGA) and eight POLLEN1LELAT52 motifs (AGAAA) (*Muschietti et al., 1994*; *Rogers et al., 2001*), and a Q element (TGACCT), which shows enhancer-like activity for the pollen-specific expression of maize (*Zea mays* L.) *ZM13* gene (*Hamilton, Schwarz & Mascarenhas, 1998*).

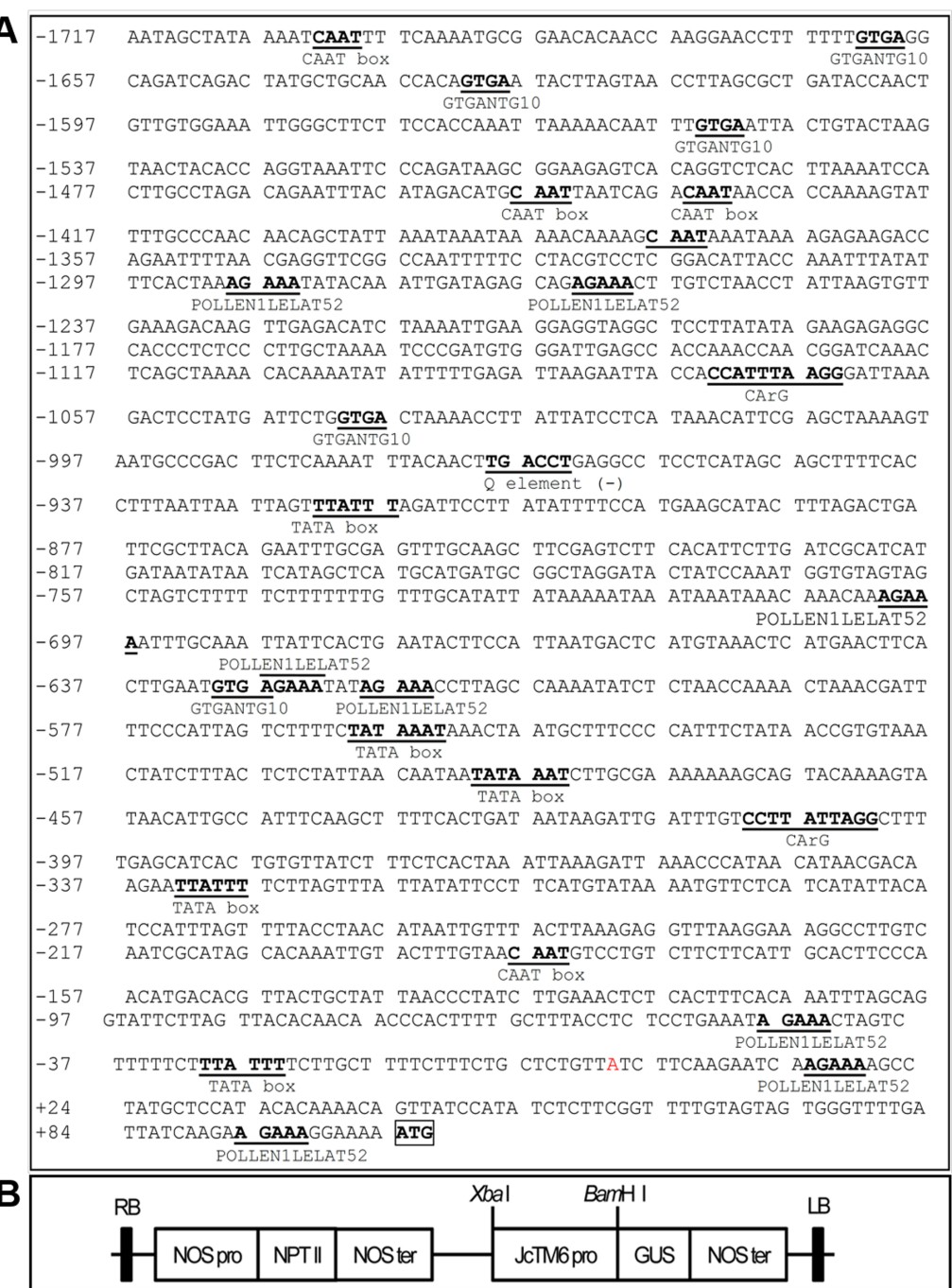

**Figure 3 JcTM6 promoter sequence and promoter-reporter gene construct.** (A) The nucleotide sequence of the *JcTM6* promoter. The transcription start site (+1) is in red. The start codon ATG is in bold and boxed. Putative regulatory elements on both strands are shown in bold and underlined. (B) A schematic of the T-DNA regions of the *JcTM6:GUS* binary vector used for transformation.

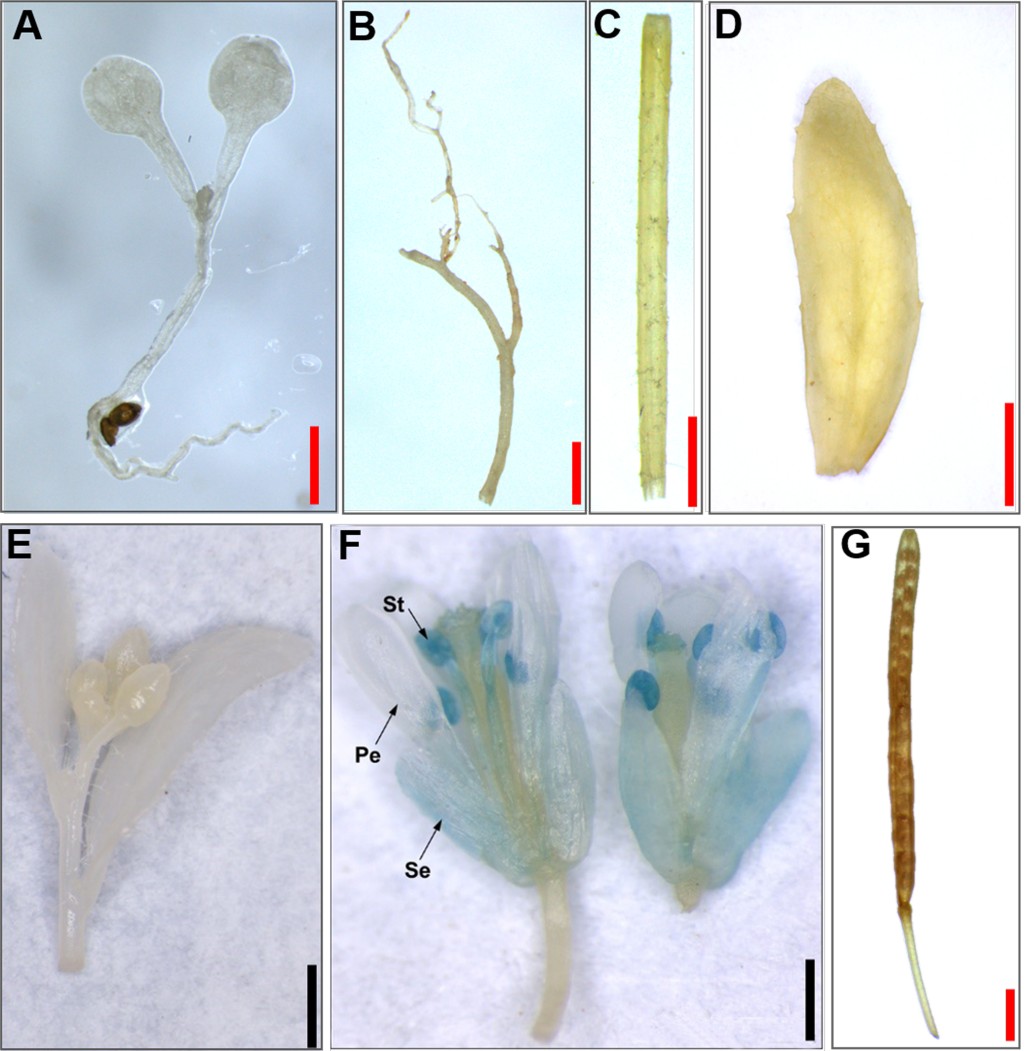

**Figure 4  Histochemical GUS staining of transgenic *Arabidopsis* harboring the *JcTM6*:*GUS* fusion.**
(A) Ten-day-old seedlings, (B) roots, (C) stems, (D) leaves, (E) inflorescence buds, (F) open flowers, (G)
green siliques. Pe, petals; Se, sepals; St, stamens. Red bars = one mm, black bars = two mm.

## Activity of the *JcTM6* promoter in *Arabidopsis*

To detect the activity of *JcTM6* promoter, a *JcTM6* promoter-GUS fusion construct
(Fig. 3B) was expressed in *Arabidopsis*, and GUS staining was monitored in homozygous
T3 plants (Fig. 4). No GUS staining was observed in 10-day-old *Arabidopsis* seedlings
(Fig. 4A). Among the five tissues of adult plants examined (including roots, stems, leaves,
flowers, and green siliques), GUS staining was detected only in flowers (Figs. 4B–4G).
Among all floral organs, GUS staining intensity was the strongest in stamens, followed
by sepals and petals, with faint staining in carpels (Fig. S1). Based on the results of GUS
staining, we conclude that the *JcTM6* promoter functions as a flower-specific promoter in
*Arabidopsis*.

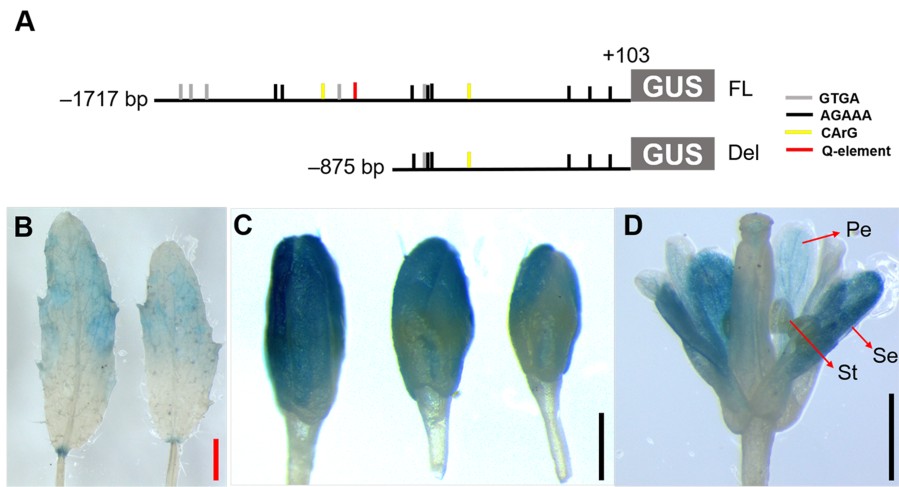

**Figure 5** **Histochemical GUS staining of transgenic *Arabidopsis* harboring the *JcTM6* deletion.** (A) Schematic representation of *JcTM6* promoter deletion. FL, full length *JcTM6* promoter, Del, deletion. GTGA: GTGANTG10 motif (gray vertical bars), AGAAA: POLLEN1LELAT52 motif (black vertical bars), CArG box: CWWWWWWWWG (yellow vertical bars), Q-element: TGACCT (red vertical bar). (B) young leaves, (C) flower buds, (D) flowers. Pe, petals; Se, sepals; St, stamens. Red bar = one mm, black bars = 0.5 mm.

## Deletion analysis of the *JcTM6* promoter

To analyze the region essential for flower-specific activity of the *JcTM6* promoter, we carried out a deletion analysis. A deletion variant of the *JcTM6* promoter lacking the region from −1,717 to −876 bp was fused to the *GUS* gene and transformed into *Arabidopsis* (Fig. 5A). Compared with the full-length *JcTM6* promoter, the deletion was not only active in flowers but also in young leaves (Fig. 5B). Moreover, the deletion showed no promoter activity in stamens but increased activity in sepals and petals (Fig. 5C and 5D). These results indicate that the region from −1,717 to −876 bp is critical for *JcTM6* promoter activity in stamens and inhibition of promoter activity in young leaves, sepals, and petals.

### *JcTM6:AtIPT4* transgenic *Arabidopsis* produced large flowers

To further verify the floral specificity of *JcTM6* promoter, a cytokinin biosynthetic gene (*AtIPT4*) was expressed under the control of *JcTM6* promoter in *Arabidopsis*. *JcTM6:AtIPT4* vector was constructed and was transformed into *Arabidopsis* plants. A total of 25 independent *JcTM6:AtIPT4* lines were obtained. As expected, all transgenic lines showed no vegetative difference from the wild type and most of them produced larger flowers (Fig. 6). Furthermore, the development of siliques was also unaffected. To verify the morphological alteration in flowers that is caused by the transgene, we examined the expression levels of *AtIPT4* and the cytokinin signaling genes *Arabidopsis histidine kinase 2* (*AHK2*) (*Nishimura et al., 2004*) and *Arabidopsis response regulator 5* (*ARR5*) (*D'Agostino, Deruère & Kieber, 2000*) in wild type and *JcTM6:AtIPT4* transgenic plants. The expression level of *AtIPT4* in flowers of transgenic lines is significantly higher than that in wild type, whereas the *AtIPT4* expression in the leaves of transgenic plants was not different from

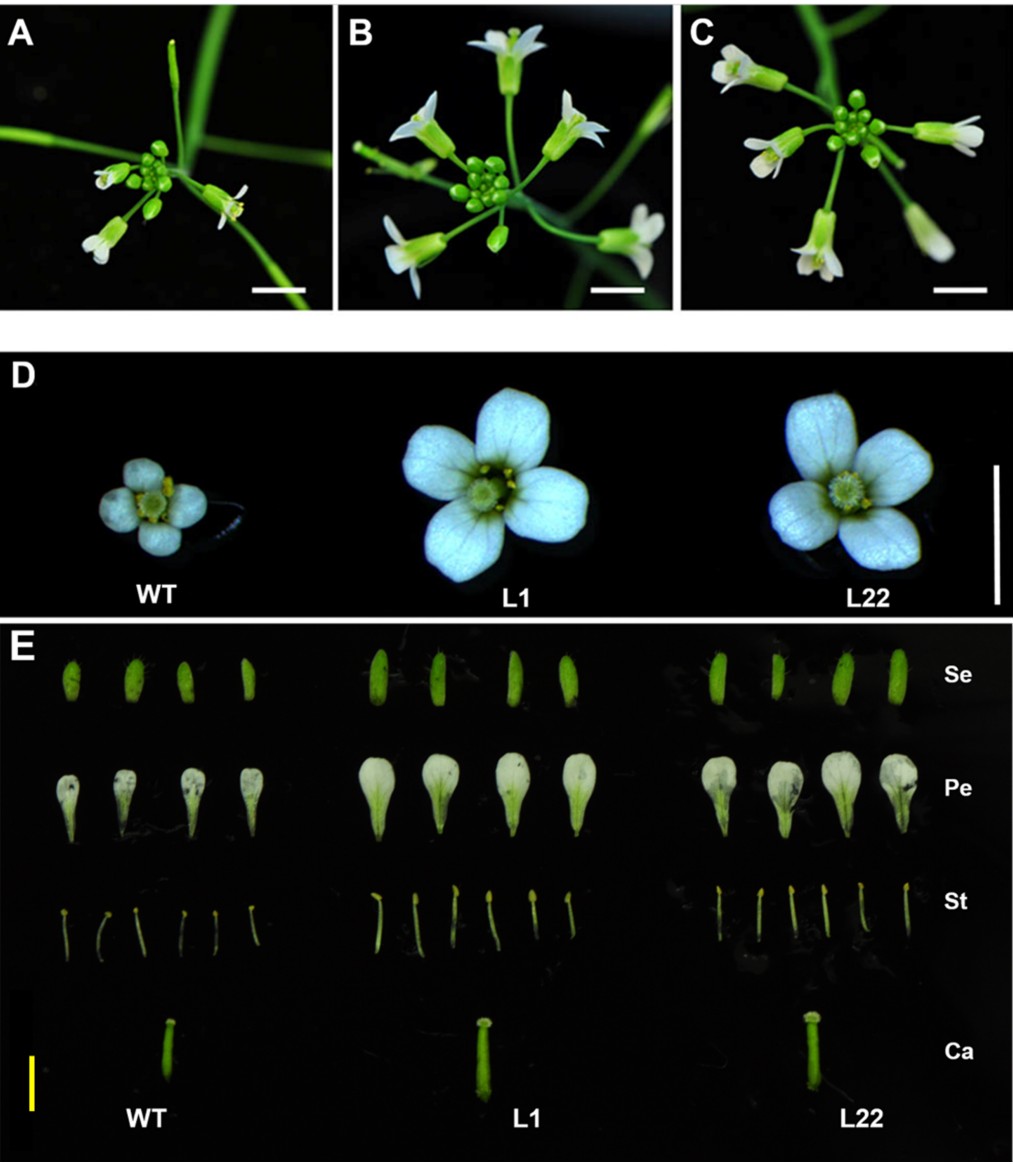

**Figure 6** **Flower size is increased in transgenic *JcTM6:AtIPT4* *Arabidopsis*.** Inflorescences of wild-type (A) and transgenic L1 (B) and L22 (C) lines. Flowers of wild-type and transgenic L1 and L22 lines (D). Dissected flowers of WT and transgenic L1 and L22 lines (E). Se, sepals; Pe, petals; St, stamens; Ca, carpels; WT, wild-type. White bars = three mm, yellow bar = two mm.

that in leaves of wile-type plants (Fig. 7A). As expected, higher expression levels of *AHK2* and *ARR5* were detected in the flowers of transgenic lines (Fig. 7B). These results indicate that the morphological alteration in flowers of *JcTM6:AtIPT4* transgenic plants is caused by the flower-specific expression of the transgene driven by the *JcTM6* promoter. *JcTM6* promoter is indeed a flower-specific promoter.

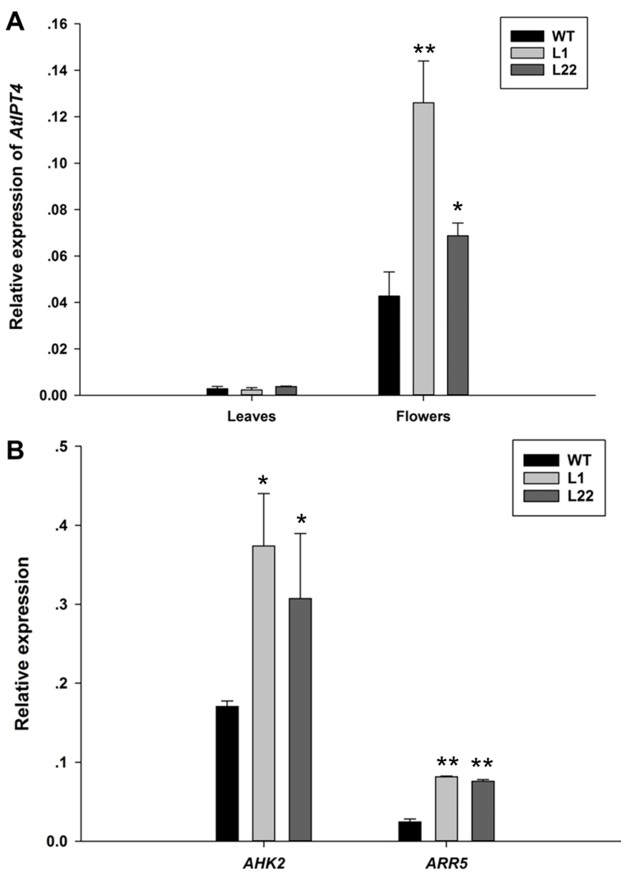

**Figure 7** **The expression analysis of *AtIPT4*, *AHK2* and *ARR5* in *JcTM6:AtIPT4* transgenic *Arabidopsis*.** (A) The expression levels of *AtIPT4* in the leaves and flowers of wild type (WT) plants and transgenic lines (L1 and L22). (B) The expression levels of *AHK2* and *ARR5* in the flowers of wild type (WT) plants and transgenic lines (L1 and L22). The values represent the means ± standard deviation ($n = 3$). Student's *t*-test was used to determine significant differences. *$p \leq 0.05$, **$p \leq 0.01$.

## DISCUSSION

*TM6* is a member of the MADS-box gene family, which belongs to the *paleoAP3* lineage (*Pnueli et al., 1991*; *Rijpkema et al., 2006*; *Wu et al., 2011*). In tomato (*Solanum lycopersicum*) and petunia, *TM6* functions as a class B gene that plays an essential role in stamen development, although it is mainly expressed in whorls 3 and 4, similar to that of a class C gene (*Martino et al., 2006*; *Rijpkema et al., 2006*). In trioecious papaya (*Carica papaya*) plants, which produce male, female, and hermaphrodite flowers, two *TM6* genes were isolated previously (*CpTM6-1* and *CpTM6-2*). Both genes are predominantly expressed in the petals of all sex types and stamens of hermaphrodite and male flowers, although *CpTM6-2* is also expressed in leaves (*Ackerman et al., 2008*). In this study, we identified *JcTM6* as a flower-specific gene in *Jatropha*, with high expression in female and male flowers (Fig. 2). Similar to *CpTM6-1*, the *JcTM6* gene showed high expression in the petals of female and male flowers and stamens of male flowers. Because *JcTM6* showed

flower-specific expression, we isolated its upstream region from *Jatropha* genomic DNA and analyzed its activity in *Arabidopsis* by GUS staining.

In transgenic *Arabidopsis*, GUS staining showed that the *JcTM6* promoter was active only in flowers (Fig. 4), suggesting that the *JcTM6* promoter is a flower-specific promoter. *AtIPT4* is a cytokinin biosynthesis gene encoding ATP/ADP isopentenyltransferase. The expression of this gene under the control of *AP1* promoter results in the alterations in flower number and organs (*Li et al., 2010*). However, the *AtIPT4* driven by *JcTM6* promoter only gave rise to the changes in flower organs (Fig. 6), indicating that *JcTM6* promoter is active at the late stage of flower development rather than floral meristem. This activity is consistent with the expression pattern of the *JcTM6* gene in *Jatropha*. Recently, *Ming et al. (2020)* showed that *JcTM6* promoter has a high activity in female flowers of *Jatropha*, suggesting that *JcTM6* promoter can drive flower-specific expression of transgenes in different plant species.

When the 842-bp fragment of the *JcTM6* promoter (−1,717 to −876 bp) was deleted, the promoter was not only active in flowers but also in young leaves (Fig. 5B). We found that the deleted region contained one of the two CArG box motifs, which are very important for mediating the regulatory effect of MADS-box transcription factors (*Dolan & Fields, 1991*; *Treisman, 1992*). In *Jatropha*, a fragment of the *JcAP1* promoter (from −1,313 to −1,057 bp), which contains a CArG box motif, is required for promoter activity in inflorescence buds (*Tao et al., 2016*). The *Arabidopsis AP3* promoter contains three CArG boxes: CArG1 is essential for *AP3* promoter activity at all stages of flowering; CArG2 is critical for *AP3* expression in petals, and CArG3 represents the binding site of a transcription factor that represses the activity of *AP3* promoter during early floral stages (*Tilly, Allen & Jack, 1998*). Therefore, we propose that the CArG box motif in *JcTM6* promoter plays an important role in conferring floral-specific activity in transgenic plants.

Among the floral organs, stamens exhibited the highest activity of *JcTM6* promoter (Fig. 4F). This expression pattern could be regulated by pollen-specific elements contained in this promoter, including five GTGA and eight AGAAA motifs. The GTGA motif is critical for the expression of *g10* promoter in tobacco pollen because mutation of the GTGA motif reduced *g10* promoter activity in pollen (*Rogers et al., 2001*). The AGAAA motif, which was identified in the tomato late-stage pollen-specific *LAT52* promoter, is necessary for promoter activity during pollen maturation (*Bate & Twell, 1998*). In potato (*Solanum tuberosum L.*), the GTGA and AGAAA motifs present in the promoter of *SBgLR*, a pollen-specific gene, are critical for high-level gene expression in pollen (*Lang et al., 2008*). In the current study, deletion of an 842-bp fragment of the *JcTM6* promoter, containing four GTGA and two AGAAA motifs, abolished promoter activity in stamens (Fig. 5D). We assumed that these motifs are essential for the activity of the *JcTM6* promoter in stamens. Given the importance of CArG box motifs, it is possible that the GTGA and AGAAA motifs cooperate with the CArG box to regulate *JcTM6* promoter activity in stamens. In addition, although the deleted region contained six AGAAA motifs, these motifs do not seem to be required for *JcTM6* promoter activity in stamens. Furthermore, the deleted region also contained a 6-bp quantitative element (Q-element), which plays an enhancer-like role (*Hamilton, Schwarz & Mascarenhas, 1998*). In maize, deletion of the Q-element from

the pollen-specific *ZM13* promoter reduced the promoter activity by 10-fold (*Hamilton et al., 2000*). Deletion of the Q-element probably also contributed to the loss of *JcTM6* promoter activity in stamens in this study (Fig. 5D). In addition, the deletion variant of the *JcTM6* promoter exhibited increased activity in sepals and petals (Fig. 5C and 5D), indicating the presence of potential negative elements in the deleted region, which inhibit promoter activity in sepals and petals. By the deletion analysis of the *JcTM6* promoter, we demonstrate the combination of these elements are of great importance to the promoter activity in the flowers, and detailed studies of the functions of these elements will be conducted in the future.

## CONCLUSIONS

Floral-specific promoters play crucial roles in genetic modification of flowering characteristics. In this study, a 1.8-kb *JcTM6* promoter fragment was isolated from *Jatropha* and characterized as a flower-specific promoter in transgenic *Arabidopsis* plants. When the region from −1,717 to −876 bp in the *JcTM6* promoter was deleted, the promoter lost its flower-specific activity and gained activity in young leaves. Our results suggest that the *JcTM6* promoter could be used to drive flower-specific expression of transgenes in plants.

## ACKNOWLEDGEMENTS

The authors gratefully acknowledge the Central Laboratory of the Xishuangbanna Tropical Botanical Garden for providing the research facilities.

### Funding

This work was supported by the Natural Science Foundation of Yunnan Province (2016FB048), the National Natural Science Foundation of China (31370595) and the Programme of the Chinese Academy of Sciences (kfj-brsn-2018-6-008 and 2017XTBG-T02). The funders had no role in study design, data collection and analysis, decision to publish, or preparation of the manuscript.

### Grant Disclosures

The following grant information was disclosed by the authors:
Natural Science Foundation of Yunnan Province: 2016FB048.
National Natural Science Foundation of China: 31370595.
Programme of the Chinese Academy of Sciences:  kfj-brsn-2018-6-008,  2017XTBG-T02.

### Competing Interests

The authors declare there are no competing interests.

## Author Contributions

- Jing-Xian Wang performed the experiments, analyzed the data, prepared figures and/or tables, authored or reviewed drafts of the paper, and approved the final draft.
- Xin Ming performed the experiments, analyzed the data, prepared figures and/or tables, and approved the final draft.
- Yan-Bin Tao conceived and designed the experiments, analyzed the data, authored or reviewed drafts of the paper, and approved the final draft.
- Zeng-Fu Xu conceived and designed the experiments, authored or reviewed drafts of the paper, and approved the final draft.

## DNA Deposition

The following information was supplied regarding the deposition of DNA sequences:

The JcTM6 gene and promoter are available at GenBank: MN820724 and MN044579.

## Data Availability

The raw data are available as a Supplementary Files.

## Supplemental Information

Supplemental information for this article can be found online at http://dx.doi.org/10.7717/peerj.9827#supplemental-information.

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
