# Peer review of "Jatropha curcas ortholog of tomato MADS-box gene 6 (JcTM6) promoter exhibits floral-specific activity in Arabidopsis thaliana"

_PeerJ, doi:10.7717/peerj.9827_

## Round 0.1 · original submission · Major Revisions

Two contradictory recommendations were suggested by reviewers. Please take all comments from reviewers into consideration to make revision.

Reviewer 1 ·

Basic reporting

Physic nut (Jatropha curcas L.) is an important biodiesel resource, although its low seed yield significantly limits its potential. The authors of this study sought to identify a Jatropha curcas promoter that would likely be involved in flower yield improvement, it’s an interesting project.

Experimental design

The experimental design is good, but there are still some experiments and results should be provided.

Validity of the findings

This study isolate a JcTM6 promoter in Jatropha curcas, which might be useful for the jatropha breeding in the furture studies.

Additional comments

However, there are still some comments for the authors to substantively revise the manuscript.

1. One of the major issue is the approach adopted to analyze the role of JcTM6 promoter in Arabidopsis. The authors expressed JcTM6 promoter and introduce this construct into Wild-type Arabidopsis plants. Then all the comparison made between wild-type Arabidopsis plants and wild-type plants that over expressed JcTM6 promoter. As we all know, the Arabidopsis thaliana is a model plant, and a frequent but inappropriate or inconclusive approach to isolate the function of the heterologous gene.

2. The function of JcTM6 is not clarified in this study. Only the expression pattern of JcTM6 is not enough to illustrate its roles.

3. Based on the manuscript, the fragment of the JcTM6 promoter is about 1.8-kb, from–1717 to +103 bp. The rough result is that why only the 842-bp fragment of the JcTM6 promoter (–1,717 to –876 bp) was deleted to identify the function of JcTM6. The author shows that this deleted region contained one of the two CArG box motifs, but what about the other one. I noticed the Arabidopsis AP3 promoter contains three CArG boxes with different functions (Line 215-218). Generally, a series of deletions of the JcTM6 promoter should be generated to analyze the regulatory effect of different regions of the promoter, and then the vital region could be accurately confirmed. I think more interesting and useful results would be obtained.

4. I noticed that JcTM6 promoter was used to drive the AtIPT4 gene. I think the Figure 6 is rough to present the function of JcTM6 promoter. It’s not clear that the different morphological results were caused by the JcTM6 or AtIPT4 in Arabidopsis thaliana. Moreover, the sole morphological results provided do not exclude a possible silencing of the endogenous transcript. So, I suggest that, at least, the cytokinin contents must be investigated to confirm the results, and not only the morphological observations. The characters of floral organs and the expression abundance of the genes associated with cytokinin synthesis and transduction pathways should also be observed between the wild type and transgenic plants.

5. I did not check whether the references were all mentioned.

I would suggest the authors should make substantial and in-depth revisions to the manuscript. The conclusions in the manuscript are based on more data than the data presented in the current manuscript, and then the authors should restate their discussion and conclusions.

Reviewer 2 ·

Basic reporting

The article untitled ‘Jatropha curcas ortholog of tomato MADS-box gene 6 (JcTM6) promoter exhibits floral-specific activity in Arabidopsis thaliana’ is well written and clear. English is correct and easy to understand. I have only a small correction on line 190 ‘TM6 functions as a class B gene that play an essential role in stamen development’ and on line 193 ‘Both genes are predominantly …..’

The literature references are adequate. Nevertheless, there is a typing mistake on line 145 and 359 ‘Meyerowitz’ instead of ‘Meyerowltzt’

The context is clearly explained in the Introduction. However, I suggest you add some information about the flowering and reproduction of Jatropha (lined 60-61). For example, mention that Jatropha is a monoecious plant species. It would help the reader to understand the importance of male to female flower ratio. It would also clearly state why you investigate the gene expression in male and female flowers.

The structure of the article is clear and the figures and tables are clear and adequate. The raw data are shared in the supplemental data

Experimental design

The proposed article corresponds to the scopes of the journal. The research questions are well defined in the Introduction. The experimental design is relevant to answer the research questions. The methods are clearly described but the number of replicates is sometimes missing. Please provide the number of biological replicates and repetitions that were used for the qRT-PCR analysis (line 104) and the GUS staining assays (line 130).

Regarding the deletion analysis, it is not clearly stated why you choose to remove this part of the promoter and not another neither why you analyze only one deletion and not several to understand more clearly the role of the different parts of the promoter. Please justify your strategy either in the 'material and methods' or the 'result' sections.

Validity of the findings

The results are well presented and raw data are available. The conclusions are well stated and answer the research question.
Regarding the phenotype of the JcTM6::AtIPT4 transgenic Arabidopsis flowers, were all floral organs affected in the same way? Are the stamens and petals more affected than the other floral organs? Indeed, you show that the promoter is mainly active in the stamens and that the gene expression is higher in petals and stamens. Do you have some numerical data for flower comparison? It would help to be more precise on the observed phenotype. On the same way, you conclude from your results that the JcTM6 promoter is a flower-specific promoter. However, it is not active in all floral organ. Could we consider that it is more a stamen-specific promoter?

Additional comments

no additional comment

---

## Round 0.2 · Major Revisions

Please take comments from Reviewer 3 into consideration to make revision.

Reviewer 2 ·

Basic reporting

all my previous comments and requirements were answered in the new version of the manuscript

Experimental design

all my previous comments and requirements were answered in the new version of the manuscript

Validity of the findings

all my previous comments and requirements were answered in the new version of the manuscript

Additional comments

all my previous comments and requirements were answered in the new version of the manuscript

Reviewer 3 ·

Basic reporting

No comment

Experimental design

An aim of this study is isolation of the promoter that is specifically active in the flower. The author’s experiments are all necessary to reveal the flower-specific promoter activity of JcTM6 in Arabidopsis and the results are support their hypothesis. However, a part of the experiments is not enough.
My concern on the experimental design is the deletion analysis of the JcTM6 promoter (Figure 5). Authors searched the putative cis elements in the JcTM6 promoter using PLACE database and found that many putative sequences of the four type of cis elements were located in the promoter (Figure 3). Then, they analyzed a deletion JcTM6 promoter using Arabidopsis transgenic plants, but it is thought that this deletion analysis is very rough. Authors discussed cis elements that is required for the stamen (and petal) specific activity of the JcTM6 promoter region and proposed that a CArG motif, that is located around -1,060-bp region of the promoter, is important for the specific manner (Line 233 to 235). However, because another two motifs, GTGA and AGAAA, are also included in the deletion region between -1,717 bp and -875 bp (Figure 3 and 5), it is also possible that the GUS activity of the deletion promoter may result from the combined effect of deletion of CArG, GTGA and AGAAA. Therefore, authors should carry out the deletion promoter analyses using several types of the deletion JcTM6 promoter that have different length of the deletion regions. Alternatively, authors should analyze the promoter that include mutations or deletion in the CArG motif that is located around -1,060 bp region. These results from mentioned above analyses will be help to understand the promoter activity and regulatory mechanism of the JcTM6 promoter that is shown a flower-specific activity.

Validity of the findings

Another my concern is the JcTM6 promoter activity in petals. The qRT-PCR analysis of JcTM6 in Jatropha (Figure 2) indicated that highly amount of its transcript is detected in the petals of male and female Jatropha flowers like as in stamens of male flower. Ming et al. (2020) reported that the GUS activity of the JcTM6 promoter:GUS fusion gene was observed in the petals of female flowers in Jatropha. The JcTM6 promoter:AtIPT4 transgenic plants (Figure 6) have large petals in its flowers compared to that in wild type. There results suggested that the JcTM6 promoter is active in either Arabidopsis petals or Jatropha petals. In contrast, In the JcTM6 promoter:GUS analysis (Figure 4), authors mentioned that faintly GUS staining was observed in the Arabidopsis petals (Line 168). This result is not enough to support the possibility that the promoter is active in the petals. Considering these matters, I recommend authors that they should carry out the histochemical analysis of the promoter:GUS fusion gene in the different stages during flower development to confirm whether strong promoter activity is observed in the Arabidopsis petals.

---

## Round 0.3 · accepted · Accept

Based on recommendation by two reviewers, this manuscript can be accepted for publication

Reviewer 3 ·

Basic reporting

No comment.

Experimental design

No comment.

Validity of the findings

No comment

Additional comments

Thank you for revisions of your manuscript. I agree your all change of this manuscript to respond my concerns. I would like to recommend you one point in the "Discussion" section as follows;

Line 215 to 216:
In transgenic Arabidopsis, GUS staining showed that the JcTM6 promoter was active only in flowers (Fig. 4), suggesting that the JcTM6 promoter is a flower-specific promoter.
->
In transgenic Arabidopsis, GUS staining showed that the JcTM6 promoter was active only in flowers (Fig. 4 "and Fig. S1"), suggesting that the JcTM6 promoter is a flower-specific promoter.

It would be my pleasure if you will consider my suggestion.